# Exclusion Diets in Functional Dyspepsia

**DOI:** 10.3390/nu14102057

**Published:** 2022-05-14

**Authors:** Stefan Lucian Popa, Dinu Iuliu Dumitrascu, Cristina Pop, Teodora Surdea-Blaga, Abdulrahman Ismaiel, Giuseppe Chiarioni, Dan Lucian Dumitrascu, Vlad Dumitru Brata, Simona Grad

**Affiliations:** 12nd Medical Department, “Iuliu Hatieganu” University of Medicine and Pharmacy, 400000 Cluj-Napoca, Romania; popa.stefan@umfcluj.ro (S.L.P.); dora_blaga@yahoo.com (T.S.-B.); abdulrahman.ismaiel@yahoo.com (A.I.); ddumitrascu@umfcluj.ro (D.L.D.); costinsimona_m@yahoo.com (S.G.); 2Department of Anatomy, “Iuliu Hatieganu” University of Medicine and Pharmacy, 400006 Cluj-Napoca, Romania; 3Department of Pharmacology, Physiology, and Pathophysiology, Faculty of Pharmacy, “Iuliu Hatieganu” University of Medicine and Pharmacy, 400349 Cluj-Napoca, Romania; cristina.pop.farmacologie@gmail.com; 4Division of Gastroenterology, University of Verona, AOUI Verona, 37134 Verona, Italy; chiarioni@alice.it; 5Division of Gastroenterology and Hepatology, University of North Carolina at Chapel Hill, Chapel Hill, NC 27599-7080, USA; 6Faculty of Medicine, “Iuliu Hatieganu” University of Medicine and Pharmacy, 400000 Cluj-Napoca, Romania; brata_vlad@yahoo.com

**Keywords:** FODMAPs, diet, functional dyspepsia, functional gastrointestinal disorders, disorders of the brain–gut interaction

## Abstract

Functional dyspepsia represents one of the most common and prevalent disorders of the brain–gut interaction, with a large number of widespread risk factors being identified. With an intricate pathogenesis and symptomatology, it heavily impacts the quality of life and, due to the limited efficacy of traditional pharmacological agents, patients are likely to seek other medical and non-medical solutions to their problem. Over the last few years, significant research in this domain has emphasized the importance of various psychological therapies and nutritional recommendations. Nevertheless, a correlation has been established between functional dyspepsia and food intolerances, with more and more patients adopting different kinds of exclusion diets, leading to weight loss, restrictive eating behaviour and an imbalanced nutritional state, further negatively impacting their quality of life. Thus, in this systematic review, we aimed at analysing the impact and efficiency of certain exclusion diets undertook by patients, more precisely, the gluten-free diet and the low-FODMAP diet.

## 1. Introduction

Functional dyspepsia (FD) is defined, according to the Rome IV criteria, as any combination of the following symptoms: postprandial fullness, early satiety, epigastric pain, and epigastric burning that are severe enough to interfere with the usual activities, occurring at least three days per week over the last three months with an onset of at least six months before the presentation [1,2,3]. FD is divided into two main syndromes: postprandial distress syndrome (PDS) and epigastric pain syndrome (EPS), with possible clinical manifestations that could represent an overlap between the two [1,2,3].

FD is a common disorder of the gut–brain interaction with a prevalence of 3% in the general population [4] and certain risk factors, including smoking, non-steroidal anti-inflammatory drugs, *Helicobacter pylori* infection, acute gastroenteritis, female sex, psychological comorbidity, and psychological stress being identified [1,2,3].

Micro-inflammation, gastrointestinal infections, alterations in gastrointestinal microbiota, mucosal and immune dysfunction, hypersensitivity to certain nutrients found in food, disturbances of the brain–gut interaction leading to abnormalities of gastroduodenal motility, delayed or rapid gastric emptying, impaired gastric accommodation, visceral hypersensitivity, and psychological factors might all be involved in the intricate pathogenesis of FD [5].

The management of FD includes lifestyle [6] and dietary changes [7], beside pharmacological therapy, emphasizing the *Helicobacter pylori* eradication, together with antacids, prokinetics and neuromodulators [5]. However, the efficacy of these pharmacological therapies remains limited. Other resources are psychological therapies, i.e., cognitive behavioural therapy [8], psychotherapy [9], gut-directed hypnotherapy [10], and electrical stimulation [11].

With research establishing a connection between functional dyspepsia and food intolerances, many patients are prone to adopt various exclusion diets, further leading to weight loss, restrictive eating behaviour, unbalanced nutritional state, and poor quality of life (QoL) [12,13]. Nevertheless, unauthorized information about dietary advises, food intolerance tests and food allergies from social platforms, increase the risk of an excessive exclusion diet. We conducted a systematic review in order to analyse the effect of exclusion diets in FD patients.

## 2. Materials and Methods

This systematic review was written following the preferred reporting items for systematic reviews (PRISMA) guidelines [14], with the registered systematic review number: Inplasy protocol 202250026.

### 2.1. Data Sources and Search Strategy

The electronic databases PubMed, EMBASE, and Cochrane Library were searched without any restrictions from their inception until 1 March 2022, to identify potential observational and interventional studies. The following search string was entered for PubMed ((“Functional Dyspepsia” [Mesh]) OR (“Dyspepsia” [All Fields]) AND ((“Diet” [Mesh]) OR (“Nutrition” [All Fields]) OR (“fermentable oligosaccharides, disaccharides, monosaccharides, and polyols” [Mesh]) OR (“FODMAP” [All Fields]) OR (“Gluten” [Mesh]) OR (“Lactose” [All Fields])). Furthermore, we manually sorted the pertinent results across the three databases with the purpose of reducing results bias, as shown in Figure 1.

### 2.2. Study Selection and Eligibility Criteria

Observational studies, as well as interventional studies assessing the impact of various diets on the evolution of functional dyspepsia were eligible for inclusion. The original articles were included in the systematic review and qualitive assessment if they satisfied the following criteria: (1) Observational or interventional study, population/hospital/primary care-based; (2) FD was confirmed by Rome II, III, or IV criteria; (3) FD diagnosis based on the criteria established in each study; and (4) Studies on humans solely.

Exclusion criteria: (1) studies published in languages other than English; (2) case reports, letters, reviews, short surveys, practice guidelines, press articles, and conference abstracts/papers; (3) abstracts published without full-text or with the paper unavailable; and (4) paediatric studies.

Two investigators (C.P. and S.L.P.) evaluated the titles and abstracts, so that studies satisfying the inclusion and exclusion criteria were further assessed by reviewing the full paper, while in the case of discrepancies between the two investigators, a consensus was reached through discussion.

### 2.3. Data Extraction

We extracted the following data from the included studies: author’s name, publication year, country, total subjects, study population, FD patients, mean age, gender distribution, FD severity, treatment, follow-up duration, and the main findings. Data were extracted and entered by V.D.B. while S.L.P. reviewed the extracted data for possible inaccuracies. Any discrepancies regarding the outcome of the data extraction were resolved through discussion. Extracted data were entered in a Spreadsheet while the final data were aggregated into the presented manuscript.

### 2.4. Quality Assessment

Two investigators (V.D.B. and S.L.P.) used the National Heart, Lung and Blood Institute (NHLBI) and Newcastle–Ottawa Scale (NOS) to assess the quality of the included studies. Overall, four quality assessment tools were used. Controlled intervention studies and before-and-after studies with no control group were analysed using the NHLBI tools accordingly. The quality of case–control and cross-sectional studies was assessed using the NOS for this type of study design. Eight separate assessment forms were conducted, four for controlled intervention studies, two for case–control studies, one for before-and-after study and one for the cross-sectional study included in our systematic review.

Regarding the NHLBI quality assessment tools, the evaluation criteria were answered either by “Yes”, “No”, “CD” (cannot determine), or “NR” (not reported) upon completion of the evaluation. Subsequently, studies were graded as “Poor”, “Fair”, or “Good”. For the NOS, studies were evaluated based on the number of stars they obtained, as well as the selection, comparability, the assessment of the outcome, and statistical methods used in the study. Studies were graded by stars, receiving a number of stars from 0–9, with studies obtaining seven or more stars being considered as “Good”. Any discrepancies regarding the quality assessment between the two evaluators were further solved by discussion. The rating of the included studies did not affect their eligibility in our systematic review.

## 3. Results

FODMAPs are fermentable short chain carbohydrates frequently found in a wide range of foods. Malabsorption of FODMAPs leads to intestinal fermentation, gas generation and an increase in osmotic pressure, causing mechano- and chemoreceptor stimulation, resulting in pain, altered GI motility, flatulence, and bloating [15]. Mechanoreceptor stimulation could be decreased by reducing GI gas, osmotic load, and water content, and chemoreceptor stimulation could be decreased by a reduction in the generation of short chain fatty acids [15]. Other beneficial foods include ginger (affecting gastric motility) and rice (fully absorbed from the GI tract, generating very little quantity of gas). Rice is considered low in FODMAPs and often used as a base for a low-FODMAP diet [15]. The low-FODMAP diet has also found use in a variety of gastrointestinal diseases, such as Irritable Bowel Syndrome (IBS) and Inflammatory Bowel Disease (IBD) [16,17].

Along with FODMAPs, foods with a high concentration of gluten (wheat, other grains, processed foods) are recognized by patients with FD as triggers of symptoms. Hence, gluten free diet (GFD) is one of the dietary modifications observed in patients with FD, although the exact symptom-inducing component from wheat (the fructan or the gliadin) is not always clear [18]. During a GFD, there is also an important reduction in dietary FODMAPs, leading to interpretation biases [7].

Although lactose malabsorption and lactose intolerance account significantly for the symptoms of the patients with FD, there is limited data on the effect of a LLD in FD [19,20,21].

### 3.1. FODMAP in Patients with Functional Dyspepsia

There are several differences between Asian and European functional gastrointestinal disorders regarding disease phenotype, as defined in the Rome IV consensus, developed primarily based on studies including Caucasian patients. Differences are based on language, cultural beliefs, microbiota pain experience and definition, and rates of FD/IBS overlap [15]. However, the prevalence of FODMAPs in both European and Asian diets demonstrates that the low-FODMAP diet could be applicable to IBS and FD patients in both cultures [15].

Recent studies suggest that an increase in mucosal eosinophils, mast cells, intraepithelial cytotoxic T cells, and systemic gut-homing T cells in the duodenum could be a feature of FD. These findings suggest that immune dysfunction could characterize FD [22]. Moreover, rates of self-reported non-celiac wheat/ gluten sensitivity (NCW/GS) are higher in FD patients, that experience worsening symptoms after FODMAPs, high-fat foods, and spicy foods containing capsaicin consumption. Wheat proteins and fructan may be primarily responsible for symptoms. Although immune mechanisms that control responses to food in FD are still poorly understood, they are highly likely to be common food hypersensitivities, including non-IgE-mediated food allergy and eosinophilic esophagitis [22].

Thus, evaluating the importance of wheat (gluten and fructans) in symptom improvement has been attempted in a pilot randomized double-blind, placebo controlled, dietary crossover trial, including 11 patients with Rome III criteria FD [23]. Of all, 9 patients followed, for four weeks, a diet low in fermentable oligo-, di-, monosaccharides, and polyols (FODMAPs) and low in gluten. Nepean Dyspepsia Index was used to evaluate response to diet, and patients with >30% response (*n* = 4) randomly received ‘muesli’ bars with either placebo, gluten, or fructans. Patients for whom symptoms reduced during the diet period but presented on gluten or fructans re-administration would be diagnosed with wheat related FD. Although the gluten-free and low FODMAP diet led to a general improvement in symptoms of FD, a specific wheat related FD could not be diagnosed in this cohort [23].

Additionally, Staudacher et al. evaluated the link between low FODMAP diet and FD symptom relief [24]. The study included a total of 59 patients diagnosed with FD, however, a vast majority (81%) were also diagnosed with irritable bowel syndrome. Patients were divided into two groups: one group that received low FODMAP advice (*n* = 40) and one group that received standard dietary advice (*n* = 19). Epigastric and overall gastrointestinal symptoms were evaluated using Structured Assessment of Gastrointestinal Symptom Scale (SAGIS). Although adherence to the specific diets did not differ between groups, here was a significant reduction in both epigastric score and total symptom score in the low FODMAP group compared with the standard diet group (*p* = 0.026), resulting in a high proportion of responders in the low FODMAP diet group (*p* = 0.012) [24].

Moreover, a cross-sectional study including 2987 Iranian adults evaluated the association of a low FODMAPs diet with symptoms of uninvestigated chronic dyspepsia (UCD) and FD [25]. A validated food-frequency questionnaire was used for FODMAPs intake estimation and for gastrointestinal symptoms, a validated version of the Rome III questionnaire was applied. The use of a diet low in FODMAPs was associated with increased risk of UCD in women but not in men. Additionally, low-FODMAPs diet was associated with an increased risk of postprandial fullness and epigastric pain, but no association with early satiation [25].

Additionally, in a prospective, single-blind trial, 105 patients diagnosed with FD by Rome IV criteria were randomized into two groups low FODMAP diet (*n* = 54) and traditional diet advice groups (*n* = 51), for four weeks [26]. In a second phase, for the next eight weeks, the LFD group were re-exposed to FODMAPs. Short-Form Nepean Dyspepsia Index (SF-NDI) was used for the assessment of symptom severity and quality of life. After four weeks of diet, both groups showed significant reduction in SF-NDI symptom scores compared with baseline, without differences of response between groups. Interestingly, patients with post-prandial distress syndrome or bloating had a higher response rate to low FODMAP diet (*p* = 0.04). Both interventions improved SF-NDI quality of life scores and multivariate analysis revealed that bloating and male gender were factors predicting response to LFD [26].

Tejedor et al. assessed the non-pharmacological approach for different functional gastrointestinal disorders through a multidisciplinary research based on group-consultations [27]. One group took part in sessions where patients were informed regarding various types of diets, over the counter medications, and took part in laughter therapy sessions and relaxation techniques. The other group was provided with written information discussed with patients belonging to the first group. Most patients had functional dyspepsia combined with other form of gastrointestinal affection, such as diarrhoea (18%), IBS (26%), or distention (35%). Overall, there was an improvement in gastrointestinal symptoms in both groups, with patients from group A reporting a significant reduction in the severity of baseline symptoms [27].

### 3.2. Dyspepsia and Gluten

In genetically susceptible individuals (HLA-DQ2 or HLADQ8 positive), gluten and prolamins determine a systemic immune-mediated disorder, a gluten-sensitive enteropathy or celiac disease (CD), with these patients being recommended a GFD [28]. Celiac disease has a prevalence of 1% in the general population, with a higher prevalence in patients with dyspeptic symptoms [28,29]. Patients with dysmotility-like FD and CD share common symptoms, such as postprandial fullness or bloating. Moreover, gluten free diet has been reported to normalize a slowed orocecal transit in adult patients with active celiac disease by unknown mechanism [30]. Therefore, some authors suggested that CD could be more prevalent in patients with FD. Santolaria et al. reported enteropathy (including mild changes such as lymphocytic enteropathy) in 35.9% of patients with dysmotility-like FD [31]. Based on the response to a GFD, CD was diagnosed in 19.7% of patients with FD [31]. This prevalence was much higher than previously reported in patients with dyspepsia because the criteria used to diagnose CD were different and included CD3 lymphocytic immunophenotyping [29,31].

Some patients have wheat allergy, and in these cases wheat specific IgE antibodies are positive, or the patient develops chronic eosinophilic or lymphocytic infiltration of the gastrointestinal tract. The estimated prevalence of IgE mediated food allergy is about 0.2–1% in American paediatric population [32]. In addition to CD patients, there are a lot of individuals without CD that report symptoms after ingesting wheat, with only minimal histological changes in the intestine. This condition is called ‘nonceliac gluten sensitivity’ (NCGS), “wheat intolerance”, or “wheat sensitivity”, and is presumed to be determined by other components of wheat, such as fructans or amylase trypsin inhibitors (ATI) [33]. Wheat sensitivity is quite common in the general population, with 15% of individuals included in a study conducted by Potter et al. reporting it [33]. In the same study, among the patients with self-reported wheat sensitivity, 31.3% fulfilled Rome III criteria for FD [34].

Two studies assessed the efficacy of a GFD in patients with FD [35,36]. In both studies, responders to a GFD were rechallenged, to identify those patients with NCGS that would benefit mostly from this diet. In the study conducted by Shazbakani et al., the prevalence of NCGS in patients with FD was 6.4%, and almost three times higher in the one performed by Elli et al. [35,36]. Another study conducted by Du et al. showed that a gluten rich diet is associated with dyspeptic symptoms [37].

Shahbazkhani et al. investigated a group of 77 patients with refractory FD (RFD) [35]. These patients had persistent symptoms after Helicobacter pylori eradication, eight weeks of PPIs, four weeks of amitriptyline and domperidone. The aim of the study was to identify patients with wheat induced dyspepsia. After a six-week trial of GFD, responders entered in a gluten challenge phase with crossover, which was placebo-controlled. In total, 65% of patients with RFD did not respond to a GFD. This study showed that in a highly selected group of patients with RFD, there is a chance that one in three patients would respond to a GFD [35]. Nevertheless, among responders to the GFD, 18.5% of patients (representing 6.4% from all RFD patients) were finally classified as NCGS based on symptoms recurrence (post-prandial fullness and epigastric pain/burning) during rechallenge with gluten. These patients would be candidates for a prolonged GFD, given the clear link between symptoms and gluten ingestion [35].

Another study enrolled 140 patients with FGIDs, including 22 patients with FD [36]. Furthermore, 75% of all patients (including FD patients) responded to the GFD, with improvement of general well-being [36]. However, after one week of gluten challenge, only four patients (18%) with FD reported symptom exacerbation and were diagnosed with NCGS. The authors discussed the role of placebo effect, but also the role of other wheat and foods components (like ATI, additives or FODMAPs) in symptoms’ generation [36].

Du et al. investigated the association between gluten consumption and onset of symptoms in FD patients [37]. All patients with FD underwent upper GI endoscopy with intestinal biopsies (neither villous atrophy nor crypts hyperplasia were detected), detection of anti-transglutaminase serum antibodies (normal in all subjects), but without HLA testing, to exclude CD. A gluten consumption pattern was estimated using a questionnaire that assessed frequency and quantity. The study included 101 FD patients and 31 asymptomatic controls. Patients with FD had higher scores for wheat consumption (both frequency and average quantity) compared to controls, and the scores corelated with the frequency of early satiety. Additionally, 21% of FD patients had duodenal inflammation (lymphocytic duodenosis ≥25 intraepithelial lymphocytes (IELs)/100 enterocytes), and 3% had ≥40 IELs/100 enterocytes, in duodenum corresponding to Marsh type I. However, Marsh type I histology was not considered suggestive for CD in this study. As GFD and rechallenge were not tested in this study, patients with NCGS were not identified [37].

If wheat sensitivity is proven, physicians could recommend the diet. Unfortunately, a lot of patients with FD follow these restrictive and expensive diets for years, based on personal experience or anecdotal information, in the absence of a clear proof that they have wheat sensitivity. GFD is not validated for use in FD patients. Patients should be warned that prolonged GFD can lead to several nutritional deficiencies, such as vitamin D, B12, iron, zinc, or magnesium [38]. In addition, restrictive diets would also increase hypervigilance and anxiety toward that food, contributing to symptoms anticipation [7].

The quality assessment of the included studies is outlined in Appendix A. Overall, six studies were rated as “Good” [24,25,26,35,36,37] and two studies were regarded as “Fair” [23,27] quality. All studies considered had adequate statistical methods, correctly assessing outcomes using valid and reliable measures. When it comes to controlled intervention studies, all four of them had proper randomization of participants, with patients and providers being blinded to the participants’ group assignment [23,26,35,36].

Both case–control studies had adequate definitions, representativeness of cases, selection and definition of controls, and properly assessed the exposure and compared the two groups using proper statistical methods [24,37]. We have also included one before-and-after study with no control group [27] and one cross-sectional one in our systematic review, whose quality was assessed using the NHLBI Quality Assessment and NOS tools, being rated as “Fair” and “Good”, respectively [25].

## 4. Discussion

FD is a prevalent disorder and accounts for a large number of consults for dyspeptic symptoms at gastroenterology practice [39]. FD can present with a complex clinical scenario including epigastric pain, early satiety, and post-prandial discomfort (Rome Criteria). Management of FD may be disappointing for both physicians and patients for there is a lack of available treatments with established efficacy [39]. Proton pump inhibitors (PPI), psychotropic medications, prokinetics and herbal remedies have all been reported to improve symptoms in subgroups of patients [39]. However, drug related side-effects are common and large sized randomized controlled trials are lacking [39]. Moreover, the recently endorsed European Guidelines for FD concluded that only PPI therapy could be considered effective to treat FD with alternative treatments not gaining consensus support [5].

Recent studies have associated FD with duodenal eosinophilia, suggesting that it is driven by an environmental allergen, such as non-coeliac gluten or wheat sensitivity [22]. A subset of patients with FD experience triggers exclusively related to meals, defined as the post-prandial distress syndrome in the Rome IV guidelines [15]. There is significant overlap of symptoms and implicated pathogenic factors with irritable bowel syndrome (IBS), another common disorder of gut–brain interaction, and a great proportion of patients has both diagnoses. The introduction of the low-FODMAP diet has represented an important turning point for IBS and it could be for FD as well [15,39]. Moreover, it would be appealing to prescribe a low-risk intervention to improve refractory dyspeptic symptoms.

Although most of FD patients indicate that their symptoms are triggered by nutrient ingestion, there is a lack of controlled dietary intervention studies. Nevertheless, the ingestion of certain nutrients and their effect in triggering gastrointestinal symptoms can be assessed using the Food Frequency Questionnaire (FFQ), research revealing certain foods that generate or exacerbate dyspeptic symptoms: fried foods, pastries, pickles, spices, and oranges [40,41,42]. Additionally, a high fat intake has been incriminated for causing symptoms in up to 80% of patients suffering from FD, with other products such as fruit juices, certain vegetables (onions, cabbage), red meat, coffee, alcohol, milk and dairy products, wheat (pasta, bread), and sweets also causing similar symptoms [40,41,42,43]. Table 1 illustrates the relationship between certain foods and dyspeptic symptoms [12,40,41,42,43].

Although many studies have shown that food represents an important trigger in the symptomatology of patients with FD [44,45,46,47], the question whether alterations in temperature, content, or timing of meals in FD contributes to this triggering effect still persists. A few studies reported intake of a lower number of meals in FD patients compared to controls, in some cases with a tendency for more snacks between meals [12,48,49]. Reduced fat intake has been reported in FD, as well as a reduced carbohydrate intake in a mixed FD/IBS population [12,40]. Nevertheless, a recent systematic review of 16 studies failed to show a consistent link between symptoms and dietary intake [50]. Thus, there is not enough data to confirm that dietary habits induce symptoms in FD patients, but patients are likely to have adapted their food intake patterns in an attempt to decrease symptom occurrence and severity. According to the consensus of European Society for Neuro-gastroenterology and Motility (ESNM), dietary adjustment improves symptoms in FD, although this statement is not endorsed [5].

Nevertheless, with patients suffering from FD being prone to adopting certain exclusion diets, they should be carefully advised before taking such a step and made aware of the negative impact it might have on their quality of life, given the possibility of not influencing or ameliorating their symptomatology, leading to malnutrition or abnormal dietary habits [12,13].

The main limitations of our review were the insufficient number of patients included in each study and the methodological bias represented by various types of therapies followed simultaneously.

## 5. Conclusions

Functional dyspepsia is a prevalent disorder of the gut brain interaction, which might be sensitive to diet advise. However, diverse attempts have been carried out in order to establish useful dietary recommendations for FD patients without gaining consistent results. Exclusion diets have not proved to be uniformly useful, as emphasized by our review. Practitioners should elaborate individual recommendations, according to real or perceived intolerances and to their preferences and believes. Dietary recommendations should not be dogmatic and should be based on individual tolerance. It is reasonable to advise frequent small- size meals and avoid high-fat food.

## Figures and Tables

**Figure 1 nutrients-14-02057-f001:**
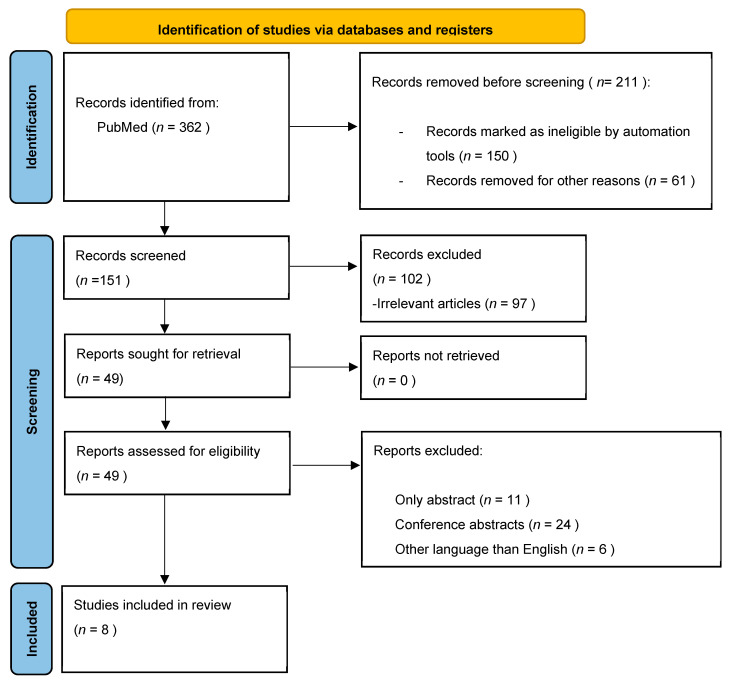
PRISMA flow diagram for study selection.

**Table 1 nutrients-14-02057-t001:** Dyspeptic symptoms and certain foods triggering them.

Symptoms	Type of Food
Early satiety	Red meat, bananas, bread, cakes, pasta, sausages, fried foods, beans, onions, mayonnaise, milk, chocolate, eggs, sweets, oranges
Bloating	Soft drinks, onions, beans, bananas
Epigastric pain Epigastric burning	Cheese, Coffee, onion, pepper, milk, chocolate, pineapple

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
