# Peer review of "Exclusion Diets in Functional Dyspepsia"

_nutrients, 2022, doi:10.3390/nu14102057_

Round 1

Reviewer 1 Report

This revised manuscript has been revised by considering the comments. Now this manuscript appears acceptable for publication without changes any more.

Reviewer 2 Report

The authors have met the review concerns previously identified. 

This manuscript is a resubmission of an earlier submission. The following is a list of the peer review reports and author responses from that submission.

Round 1

Reviewer 1 Report

The presentation of this review is confused, especially the results section which contains a lot of discussion The pathogenesis of functional dyspepsia is unknown-many people assume it is a disorder of the gut-brain axis. There are lots of papers on the involvement or not of other mechanisms-for which the authors do not give a balanced overview (bloating is often related to diaphragmatic movement). In particular it is not clear why a condition of the upper GI tract should be influenced by a lowFODMAP diet which influenced predominantly colonic microflora. Wheat allergy is not implied by having IgE antibodies. In many ways this is very hard to do and probably should be avoided due to the controversy over potential mechanisms. The review should be about benefit or not of a dietary intervention and principally limited to this. The introduction does not mention how diet could influence dyspepsia. What is food intolerance? I am quite surprised by the small number of studies-there is little criticism of these studies design. I cannot see how the participants could have been blinded to intervention. There are studies of food challenges given via NG tubes that could be mentioned. The follow up period for most studies is inadequate. Overall the picture for the use of diet to treat dyspepsia is not clear, let alone the mechanism. The quality of studies need to be improved (larger sample size, longer follow up-at least 1 year)

Reviewer 2 Report

The authors of the manuscript "Exclusion Diets in Functional Dyspepsia" is a very well written manuscript in the field of Functional Dyspepsia which is not being studied a lot. Since FD is a disorder of gut brain its more relevant to study the field.  The manuscript could lead to new therapeutic agents other than PPI which is more likely used.

There is only one question to be answered:

In the clinical studies used by the authors where the studies on same ethenic group or on different ethenic groups  and from which parts of the world where  the studies choose.  If the authors can clarify.

Reviewer 3 Report

The purpose of this review is interesting as there is a concern about people following unhealthy diets for various functional GI diseases on their own without good knowledge of the possible effects. The purpose appears to be to analyze the effect of exclusion diets in functional dyspepsia (FD). One is led to believe that the review might include effects such as unintended weight loss, restrictive eating behaviors, unbalanced nutritional status, and quality of life.

It is noted under 2.2 Study Selection and Eligibility Criteria that only observational studies were to be included and clearly states that experimental studies would be excluded, however, controlled intervention studies and pre- and post-intervention studies were included. In the results and discussion, it appears as though experimental studies were included unless a definition other than those standard in the scientific world were used. If so, it needs to be clarified as to the studies used.

A consort diagram was used which is helpful but is a bit unclear without explanation. For example, how were records marked as ineligible through automation or for other reasons? It isn’t clear how records were screened given the results and the exclusion / inclusion criteria.

The coding and the instruments used are well described.

The results are not clear and some of the information appears as background and should be moved to the introduction such as the first four paragraphs. A summary of the types of studies and findings would be helpful.  The results explain the findings of studies that explored results of different exclusionary diets, especially on the FD symptoms. I did not see results or discussion of the effects such as the unbalanced nutritional status or quality of life related to diet rather than symptoms.

Note that the prevalence in the discussion is different from that noted in the introduction. The discussion has a lot of background information rather than a comparison of the results to other reviews or studies. The conclusions that diets have not be uniformly helpful and the recommendation for small – sized meals and low fat are not substantiated by the results.

In general, the purpose statement, methodology, results and conclusion do not match or follow from one another. It is not clear what the purpose was given the results and discussion or what was actually included and if the methods were followed. The conclusions may be realistic practice suggestions, but do not follow from the review.

Reviewer 4 Report

This systematic review by Popa et al. demonstrated the exclusion diet in functional dyspepsia. The point of view is very interesting. There are some issues that should be addressed

  1. There are no figures in this manuscript. I think it is better to make some figures for the understanding of readers on the relationship between gluten-free diet, low-FODMAP and the symptom generation of functional dyspepsia.

  1. Page 4, line 142; The full spelling of LLD should be written.